# Evolvability in the Cephalothoracic Structural Complexity of *Aegla araucaniensis* (Crustacea: Decapoda) Determined by a Developmental System with Low Covariational Constraint

**DOI:** 10.3390/biology11070958

**Published:** 2022-06-24

**Authors:** Erwin M. Barría, Hugo A. Benítez, Cristián E. Hernández

**Affiliations:** 1Departamento de Ciencias Básicas, Facultad de Ciencias, Universidad Santo Tomás, Los Carrera 753, Osorno 5290000, Chile; 2Centro de Investigación e Innovación para el Cambio Climático, Facultad de Ciencias, Universidad Santo Tomás, Santiago 8370003, Chile; 3Laboratorio de Ecología Evolutiva y Filoinformática, Departamento de Zoología, Facultad de Ciencias Naturales y Oceanográficas, Universidad de Concepción, Casilla 160-C, Concepción 4070386, Chile; cristianhernand@udec.cl; 4Programa de Doctorado en Sistemática y Biodiversidad, Facultad de Ciencias Naturales y Oceanográficas, Universidad de Concepción, Casilla 160-C, Concepción 4070386, Chile; 5Laboratorio de Ecología y Morfometría Evolutiva, Centro de Investigación de Estudios Avanzados del Maule, Universidad Católica del Maule, Talca 3466706, Chile; hbenitez@ucm.cl; 6Centro de Investigación en Recursos Naturales y Sustentabilidad (CIRENYS), Universidad Bernardo O’Higgins, Avenida Viel 1497, Santiago 8370993, Chile; 7Universidad Católica de Santa María, Arequipa 04000, Peru

**Keywords:** covariation, palimpsest model, developmental pathway, asymmetry, canalization

## Abstract

**Simple Summary:**

The origin of complex morphological structures is explained mainly by direct pathways fusing adjacent modules, while the independent effect of parallel pathways acting on different areas of a morphogenetic field is less well-known. The palimpsest model that explains the cephalothoracic structural complexity of decapod crustaceans is composed of two hox-regulatory parallel pathways that tagmatize the anterior metameres early, followed by a direct pathway that fuses the tagmata forming the developmental modules. The cephalothoracic geometry of *Aegla araucaniensis* shows a marked sexual dimorphism; its adaptive causes also promote dimorphic variations in the evolvability of developmental modularity. We found areas of instability in the variance of the asymmetry in both developmental modules. The direct pathway presents intermediate levels of canalization in the covariation of the developmental modules, although significantly higher in males. This low restrictive potential promotes expressions of gonadic modularity in females and agonistic modularity in males, which differ significantly from developmental modularity. The cephalothoracic palimpsest model of decapods allows studying modularity in an explicit evo–devo context.

**Abstract:**

The integration of complex structures is proportional to the intensity of the structural fusion; its consequences are better known than the covariational effects under less restrictive mechanisms. The synthesis of a palimpsest model based on two early parallel pathways and a later direct pathway explains the cephalothoracic complexity of decapod crustaceans. Using this model, we tested the evolvability of the developmental modularity in *Aegla araucaniensis*, an anomuran crab with an evident adaptive sexual dimorphism. The asymmetric patterns found on the landmark configurations suggest independent perturbations of the parallel pathways in each module and a stable asymmetry variance near the fusion by canalization of the direct pathway, which was more intense in males. The greater covariational flexibility imposed by the parallel pathways promotes the expression of gonadic modularity that favors the reproductive output in females and agonistic modularity that contributes to mating success in males. Under these divergent expressions of evolvability, the smaller difference between developmental modularity and agonistic modularity in males suggests higher levels of canalization due to a relatively more intense structural fusion. We conclude that: (1) the cephalothorax of *A. araucaniensis* is an evolvable structure, where parallel pathways promote sexual disruptions in the expressions of functional modularity, which are more restricted in males, and (2) the cephalothoracic palimpsest of decapods has empirical advantages in studying the developmental causes of evolution of complex structures.

## 1. Introduction

Inferences in morphological evolution generally tend towards structuralist explanations if they are based on factors of origin and development or towards functional explanations if they respond to natural selection [1]. However, when combining both points of view, it is possible to obtain a fuller understanding of the role of the origin and development on the capacity for evolutionary change (evolvability) of the morphological traits [2,3]. A synthesis emerges from this, described in three corollaries of the developmental fundamentals of the morphological modularity and their ecological and evolutionary implications: (1) body plans are formed by subsets of internally integrated structures (modules) variably cohesive among themselves; (2) there is a pervasive relationship between higher intra-module cohesion and lower inter-module cohesion regulated by the developmental precursors (developmental modularity), and (3) such precursors tend to reduce the co-variability of the modules by canalization. Therefore, the developmental precursors (or pathways) constitute drivers of stabilizing selection, restricting the covariational expression and/or optimizing the functional performance of developmental modularity [4,5,6]. This principle is widely supported by the effect of direct developmental pathways that regulate the fusion of components in complex morphological structures, where the canalization of developmental modularity is proportional to the intensity of the structural fusion [7,8]. However, the response of developmental modularity is poorly understood when processes that generate complex structures with less cohesion between modules are involved, such as the differentiation of a morphogenetic field by the independent regulation of different parallel developmental pathways [6].

The origin of the cephalothoracic structural complexity of decapod crustaceans is explained by the early action of two parallel Hox regulatory pathways responsible for the tagmatization of the cephalon and pereon (Figure 1A). During metamorphosis or late embryogenesis in species with direct development, a change in Hox regulation occurs in the anterior metameres of the pereon that fuses both tagmata functionally (Figure 1B). The pereonic appendages, or maxillipeds, affected by this direct developmental pathway change from large locomotor to small perioral appendages highly coordinated with the functionality of cephalonic structures such as the mouth, mandible, maxillule and maxilla [9,10]. The cervical groove emerges, anatomically delimiting the developmental modules (Figure 1C). This model fits a simple palimpsest, where the effect of parallel developmental pathways, as promoters of modularity, reduce or compete with the canalization imposed by the direct developmental pathway [11]. The lower restrictive potential involved in this regulatory system suggests signals of evolvability for developmental modularity, reflected in the ability to change the covariation structure and express patterns of functional modularity with the fitness value [12,13].

This prediction was evaluated here by estimating the intensity of the expression of the direct developmental pathway and its effect on the evolvability of developmental modularity in the adult cephalothorax of *Aegla araucaniensis*, an endemic and abundant freshwater anomuran crab from South-Central Chile that has a marked adaptive sexual dimorphism in the shape of its carapace [14]. The greater cephalothoracic width towards the caudal end of females provides internal space to cover the variations in ovarian size during the gonadic cycle and favor the well-being of the embryos during external incubation under the pleon. Both functions optimize the quantity and quality of the new progeny, favoring population fitness by natural selection [15,16]. The greater anterior amplitude and prominence of the latero-frontal spines in males determine the performance of agonistic confrontations between males and copulatory behavior [17,18,19] These responses favor the individual fitness of males through mating success, which involves an important component of sexual selection [16,20,21]. We hypothesize that these divergent adaptive causes also influence the sexually dimorphic expression of functional modularity, because the direct developmental pathway is a weak driver of cephalothoracic canalization, which implies a low correlation in the asymmetry of the developmental modules and significant changes in the covariation structure that defines its modularity.

The Hox regulation model that explains the cephalothoracic structural complexity in decapod crustaceans offers important advantages as a study model in evo–devo [22]. The developmental pathways approach captures the more general features of these genetic regulation processes [23], allowing a palimpsest model formed by two sequential mechanisms of structural complexity with antagonistic covariational effects [11]. The rigidity and compartmentalization of the decapod body plan offer multiple homologies that morphologically define and delimit developmental modules [24], which is ideal to measure the modularity and infer developmental interactions from landmark covariations and correlations of bilateral asymmetry [25,26]. The intensity of the direct developmental pathway was inferred by correlating the asymmetry generated by common regulatory perturbations that coordinately deviate from the bilateral symmetry of the modules [27,28,29]. The evolvability of developmental modularity was inferred by contrasting the strength of the expression of its covariation structure and that of additional modularity models attributable to the functional and anatomical factors [26,30,31,32]. The study model and the empirical framework proposed will contribute to understanding the covariational evolution of complex phenotypes beyond the constraints imposed by a particular type of developmental factor.

## 2. Materials and Methods

We analyzed the landmark configurations from the dorsal surface of the adult cephalothorax of a sample of 55 female and 48 male of *A. araucaniensis* (Figure 2A). This sample collected in 2006 using a manual trawl net was behaviorally characterized [33] and the landmarks of this data set were digitized twice with the tpsDig2 program [34] and have been ecomorphologically analyzed in previous studies that described the capture, sexing, size, photographing, location of landmarks and fate of the individuals [14,16,33]. This configuration is composed of 33 homologous landmarks: 30 paired landmarks that describe the bilateral geometry of the anterolateral spines and the anatomical areas formed by the intersection of the dorsal lines, plus three unpaired landmarks located over the axis of bilateral symmetry (Figure 2B). This sample was subjected to a Procrustes superimposition fitted by Generalized Least Squares (GLS) and aligned on an axis of symmetry [35] to decompose the shape variations into symmetric and asymmetric components [36,37]. The variability of both components was evaluated with a Procrustes ANOVA, where the symmetric component represents the among-individual variability, the asymmetric component is explained by within-individual factors determining asymmetry (we nested sex within both effects), the interaction within-among individuals measures the fluctuations of the individual asymmetry with respect to the mean of the total asymmetry (fluctuating asymmetry) and the error term is the residual variance, which includes the effects of both replicas [36,38]. Procrustes superimposition was performed with MorphoJ 1.06d software [39], and the Procrustes ANOVA was computed in the geomorph 3.3.2 R package [40,41], which constructs distributions with Randomized Residuals in a Permutation Procedure using the RRPP 0.6.2 library [42].

The deviation from the perfect symmetry of the paired landmarks organized through the cephalocaudal axis was measured from the matrix of Procrustes coordinates of the asymmetric component. The area of the right triangle formed by the union of landmark 1 and each paired landmark was measured, and the percentage difference between the sides was used as a descriptor of the left–right asymmetry (L–R_Assym_), being a negative value if the asymmetry is skewed to the left. Longitudinal L–R_Assym_ profiles were compared within and among sexes. The deviation from the perfect symmetry of each pair of landmarks (L–R_Assym_ = 0) was compared with a Z-test performed in the BSDA1.2.0 R package [43], and the sexual differences in L–R_Assym_ were tested with paired comparisons based on a Student’s *t*-test [44], performed with the base 4.0.3 R package. The intensity of the expression of the direct developmental pathway (Figure 1B) was inferred by correlating the directional asymmetry of the shape of both developmental modules. A Mantel test was applied for each sex on the matrices of the Procrustes coordinates of the asymmetric component of both developmental modules. This analysis was performed with the ecodist 2.0.7 R package using Euclidean distances among the Procrustes coordinates and 5000 bootstrap iterations [45,46]; the differences between the sexes in the distribution of the correlation values obtained were compared with an additional paired Student’s *t*-test.

We compared the intensity of the expression of seven hypothetical models of cephalothoracic modularity. First was a simple model of uniform covariation that considers the cephalothorax as a unitary structure (Figure 2C). Three additional models of bi-modularity were considered: (1) developmental modularity proposes the maximum covariation between the landmarks within each cephalothoracic module and the minimum covariation between landmarks of different modules (Figure 2D), (2) a model of gonadic modularity suggests a greater covariation between landmarks that include the anatomical space occupied by the reproductive system of the females (Figure 2E) and (3) a model of agonistic modularity proposes more covariations among the landmarks that describe the set of frontal and lateral spiny processes (Figure 2F). Three models of anatomical modularity were included, which combined all possible subdivisions in the map of dorsal lines displayed over the thoracic module (Figure 2G–I). For this, the landmark configurations were subjected to superimposition based on the Generalized Procrustes Analysis (GPA), aligned by the centroid and optimized by the Procrustes distance [47] in tpsRelw 1.71 [48]. From the submatrices of Procrustes coordinates that summarize the parameters of each model, we estimated the Covariance Ratio (CR), a descriptor of modularity that relates to the total covariance of the elements between modules to the paired covariance of the elements within each module, and the force of the modular signal of each model was estimated as the CR effect size (Z_CR_). Since the modularity increases with the intra-module cohesion, greater modularity and a greater intensity of modular signal occur when CR tends toward zero and Z_CR_ is negative [31,32]. Distributions of Z_CR_ values for each modularity hypothesis were obtained by RRPP [49], the absolute value of the paired differences between the means of these distributions was estimated (|Ẑ_12_|) and the overlap ratio of the Z_CR_ distributions measured the statistical significance (*p*-value) of the models compared [32]. Estimations of CR, Z_CR_ and paired comparisons based on |Ẑ_12_| were performed in the geomorph 3.3.2 R package [41] and the RRPP 0.6.2 library to construct the Z_CR_ distributions [42]. We compared 19 possible additional variants of covariation from the seven modularity models proposed based on the correlations within and between modules (Appendix A). The log-likelihood support for each covariation model and the subsequent fit by AICc were estimated in the EMMLi 0.0.3 R package [27]. Since this analysis is extensive, the methodological details of this procedure are described in a Appendix A.

To assess the degree of modular structure of both developmental modularity and the best-fitting model of modularity of each sex, we applied modularity tests based on the CR values. The residuals matrix of the Procrustes coordinates of the symmetric components obtained by GLS superimposition and regressed on the centroid size [50] were partitioned according to the different modularity models. The relative frequency distributions of the CR values of the original partitions were compared to a distribution of the CR values estimated for a sample of arbitrary partitions with the same number of landmarks per module. The CR values in both distributions were obtained with 10,000 permutation rounds, changing landmarks randomly in the hypothetical modules in the original partitions and among the different modules in the arbitrary partitions. The proportion of CR values of the arbitrary partitions lower than the CR values of the original partitions was considered as an estimator of the statistical significance [31]. All procedures involved in the modularity test were performed in the geomorph 3.3.2 R package [40,41].

## 3. Results

The output of the Procrustes ANOVA indicated significant sexual dimorphism in the shape of the symmetric component and the directional asymmetry but not in the fluctuating asymmetry (Table 1). The shape longitudinal profile asymmetry showed a marked directional skew to the left in females (Figure 3A). Most of the paired landmarks had asymmetry values significantly less than zero (L–R_Assym_ < −0.20; Z-test < −2.0; *p* < 0.005). Only the landmarks of the tip of the anterolateral spines (pair 26-04), and the intersections of the branchial line and the linea aeglica lateralis (pair 18-12) were significantly skewed to the right (L–R_Assym_ > 0.33; Z-test: 3.5; *p* < 0.05), while the paired landmarks of both hepatic lobules 1 (pair 25-05) had a better fit to perfect symmetry (L–R_Assym_ = 0.014; Z-test = 0.146; *p* = 0.883). In males, 73% of the paired landmarks (11 pairs) were significantly skewed to the right (mean L–R_Assym_ from 0.37 to 2.77; Z-test from 4.92 to 20.66; *p* < 0.001; Figure 3B). The asymmetry of hepatic lobules 1 (pair 25-05), the tip of the anterolateral spine in the anterior extreme (pair 26-04) and the distal extremes of the branchial area (pair 17-13) did not differ from the perfect symmetry (L–R_Assym_ between −0.5 and 0.06; Z-test between −0.33 and 0.63; *p* > 0.05). The landmarks located at the intersections of the branchial line and the línea aeglica lateralis (pair 18-12) were the only ones with a significant asymmetry skewed to the left (mean −0.53; Z-test = −5.265; *p* < 0.0001). As a reflection of the fluctuating asymmetry, the paired landmarks of all individuals grouped did not differ significantly from perfect symmetry (mean L–R_Assym_ between −0.084 and 0.28; Z-test between 4.02 and 1.20; *p* > 0.05; Figure 3C). The asymmetry of the orbital sinuses (pair 28-02) was generally the most variable in the cephalic module, while the asymmetry of the inflection in the cervical groove (pair 31-29) was the most variable in both the thoracic module and for all cephalothorax (Figure 3). The most significant sexual differences in directional asymmetry were found on the tip of hepatic lobe 2 (pair 23-07; *t*-test = −2.470; *p* < 0.05) and the tip of hepatic lobe 3 (pair 22-08; *t*-test = −2.808; *p* < 0.01), both located at the posterior extreme of the cephalic module, and the tip of the epibranchial spine (pair 20-10; *t*-test = −2.372; *p* > 0.05), located at the anterior extreme of the thoracic module (Figure 3A,B). Both sexes expressed positive and significant correlations in the asymmetry of the developmental modules, although with intermediate magnitudes of association (Mantel’s r between 0.45 and 0.67), which was significantly higher in males (Figure 3D).

The values of the modular signal observed in the gonadic modularity of females and agonistic modularity of males were significantly higher than in the remaining bi-modularity models (including developmental modularity), which formed a cluster with intermediate Z_CR_ values, and the multi-modularity anatomical models with a significantly lower modular signal (Figure 4). A similar response was obtained applying EMMLi, where the gonadic modularity of females and agonistic modularity of males, both with similar within-module correlations, had the highest log-likelihood and AICc support (Appendix A). The developmental modularity of both sexes showed a covariance structure with a greater relative independence between modules than expected for any arbitrary partition, although the developmental modules of females (Figure 5A) showed significantly less cohesion than the developmental modules of males (Figure 5B). However, the functional modules selected in both sexes showed even more relative independence than the expression of developmental modularity. In fact, based on the CR value, the gonadic modularity in females (Figure 5C) and the gonadic modularity in males (Figure 5D) were 14.10% and 10.11% greater than their respective expressions of developmental modularity.

## 4. Discussion

The direct developmental pathway partially influences the canalization of the developmental modularity in the adult cephalothorax of *A. araucaniensis*, promoting evolvability through sexually dimorphic expressions of functional modularity. This response is consistent with the proposed hypothesis and establishes that the sex of adults (and its adaptive implications) constitutes a factor that affects not only the variation of the cephalothoracic morphology [14,16] but also the structural covariation from its developmental causes, which contributes to understanding the potential evolvability of a complex morphological structure originated by a developmental system of lower restrictive potential.

The adult cephalothorax of *A. araucaniensis* presents a particular pattern of subtle asymmetry explained by developmental determinants. Through the longitudinal cephalothoracic axis of all individuals, the cephalic module has a nucleus of greater variance of the asymmetry associated with the base of the rostral spine, at least three points of greater variance of asymmetry in the center of the thoracic module and an area of least variance in the middle of the cephalothorax. However, between the sexes, there was a significant divergence in the direction of the asymmetry, biasing it towards the left in females and towards the right in males. The maintenance of sign and magnitude observed within each sex was described as constant asymmetry by Chippindale & Palmer [51], who demonstrated its persistence during the growth of the brachyuran crab *Hemigrapsus nudus* and attributed it to the action of early developmental precursors. Our results empirically demonstrate this interpretation, based on asymmetry being a response of phenotypic instability generated by developmental perturbation [52,53]. The difference in the number of centers of instability in the head and thorax suggests the impact of sources of perturbation that independently influenced the developmental modules. In contrast, in the intermediate zone of the cephalothorax, a precursor seemed to intervene that favored the stabilization of the asymmetry variance. The palimpsest model synthesized from developmental pathways [11,23] explained this response in the cephalothorax of decapod crustaceans, where the early direct developmental pathways promote independent perturbations in each developmental module, and the latter direct developmental pathway influenced the canalization in zones adjacent to the structural fusion. The asymmetry of the developmental modules of *A. araucaniensis* also showed intermediate levels of correlation, although significantly higher in males, which indicated that the direct developmental pathway partially coordinates the covariation of the developmental modularity [53,54,55] generating a more intense cohesion in males.

The independent action of the parallel developmental pathways and the weak canalization of the direct developmental pathway reduced the cohesion between the studied developmental modules. This allowed the set of selection pressures that explain sexual dimorphism to act on the covariation structure of developmental modularity and express patterns of functional modularity relevant to the fitness of each sex, where the landmarks of the ovarian area acquire greater internal cohesion. This favors the per-offspring investment of females, while, in males, there is more orchestrated covariation in the more prominent frontal and lateral morphological attributes, which maximize the anterior muscular performance, mobility and strength of the chelipods and the reduction of damage due to confrontations [15,16,56]. Additionally, the implicit evolvability in this developmental system has a high component of natural selection that directly influences changes in expression due to the low correlation between parts, facilitating selective disruptions due to functional specialization [57]. The marked delimitation of the developmental modules in *A araucaniensis* is not only a conspicuous taxonomic synapomorphy [58] but also an anatomical imprint generated by development, which also supports the synthesis of the palimpsest model [11].

The highest modular signal and best likelihood support of gonadic modularity in females and agonistic modularity in males is a response in which two different procedures converged (Z_CR_ and EMMLi) due to the robustness of our sampling design, where the number of specimens was greater than the number of variables [30,32]. The magnitude of the modular signal, the likelihood support and the CR values in the modularity tests showed less difference between the expressions of developmental modularity and agonistic modularity in males compared to the difference between developmental modularity and gonadic modularity in females. This lower flexibility observed in the covariation structure is coherently linked to the greater intensity of cohesion exerted by the direct developmental pathway on the developmental modularity of males. Therefore, our approach based on an asymmetry and modularity analysis allows us to understand that the cephalothoracic palimpsest of *A. araucaniensis* favors an evolvable scenario of adaptive covariation but with sexually divergent levels of subtle developmental restrictions, which are more intense in males.

Sexual dimorphism in bilateral asymmetry is a morpho-adaptive pattern widely recognized in decapod crustaceans, for example, the heterochely in the cutter and crusher claws and its consequences in sexual selection, the asymmetry in the female pleopods as a reproductive character and the pleonal asymmetry of hermit crabs facilitating the habitability of gastropod shells [59,60,61,62]. We report significant sexual dimorphism in the cephalothoracic asymmetry of *A. araucaniensis*, which is explained by two factors of origin with antagonistic covariational effects whose interactions have consequences in the capacity for adaptive change of the developmental modularity. The developmentally explicit and empirically reproducible synthesis involved in the cephalothoracic structural complexity of decapod crustaceans offers advantages as a study model [22], which could improve our understanding of the ecological and evolutionary consequences of the origin and development of complex morphological structures and, particularly, explain the evolution of the body planes of decapod crustaceans from a developmental basis. For example, it may help us to understand how this developmental system has influenced carcinization, the greatest diversification of brachyuran crabs within a conservative body plan, and/or the greatest morphological disparity in the least diverse group of anomuran crabs [63,64,65].

## 5. Conclusions

Covariation models based on asymmetry and modularity allow the construction of adaptive contexts to understand the patterns of change in structural complexity from its developmental causes. Within the palimpsest model that explains the cephalothoracic structural complexity of decapod crustaceans, the adults of *A. araucaniensis* showed an important influence of parallel developmental pathways. This condition reduced the canalization potential of the direct developmental pathway with a sexually differentiated effect, where the developmental modules of the males showed a greater integration. The influence of the parallel developmental pathways was evidenced by the variability in the number of nuclei and the intensities of the variance perturbations observed in both modules through the longitudinal axis. While the canalization effect of the direct developmental pathway was reflected in the stability of the asymmetry variance observed in the area of structural fusion. The greater relative independence implicit in the expression of the developmental modularity promoted functional changes in its covariance structure. Thus, the models of gonadic and agonistic modularity displayed higher values of the modular signal and likelihood in females and males, respectively. The greater similarity between the expressions of developmental modularity and agonistic modularity of males constituted additional evidence of the greater canalization in the cephalothorax of males. Consequently, the cephalothorax of *A. araucaniensis* is a complex evolvable structure, because the effect of parallel developmental pathways promotes adaptive remodeling toward the sexually dimorphic expressions of functional modularity. The asymmetry and modularity analyses applied to the development of the cephalothorax offer a study model with theoretical value and empirical significance to understand how the mechanisms of origin influenced the ecological and evolutionary consequences of the structural complexity.

## Figures and Tables

**Figure 1 biology-11-00958-f001:**
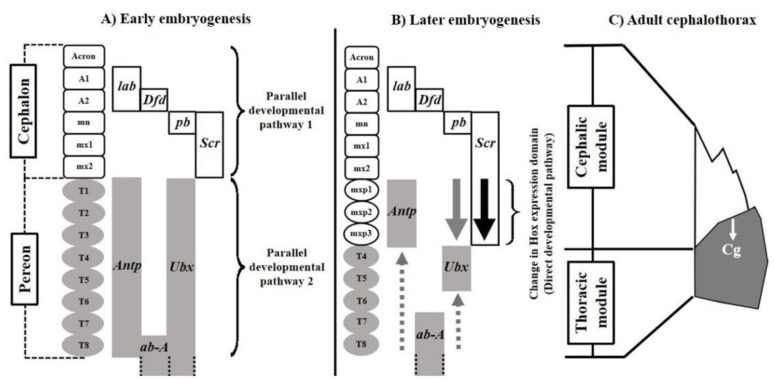
Palimpsest model explaining the origin of the structural complexity of the cephalothorax of decapod crustaceans. (**A**) Parallel developmental pathways regulating the early tagmatization of the cephalon and pereon. (**B**) Change of the Hox regulation as a direct developmental pathway promoting the fusion of the tagmata. (**C**) Structure resulting from the fusion of the developmental modules. Cg: cervical groove, lab: labial, Dfd: Deformed, pb: proboscipoidea, Scr: Sex comb reduced, Antp: Antennapedia, ab-A: abdominal-A and Ubx: Ultrabithorax. (Synthesized from Klingenberg [5,6] and Abzhanov & Kaufman [9,10]). Abbreviations for appendages: A: Antennae, mn: mandible, mx: maxillae, T: toracopodite and mxp: maxilliped.

**Figure 2 biology-11-00958-f002:**
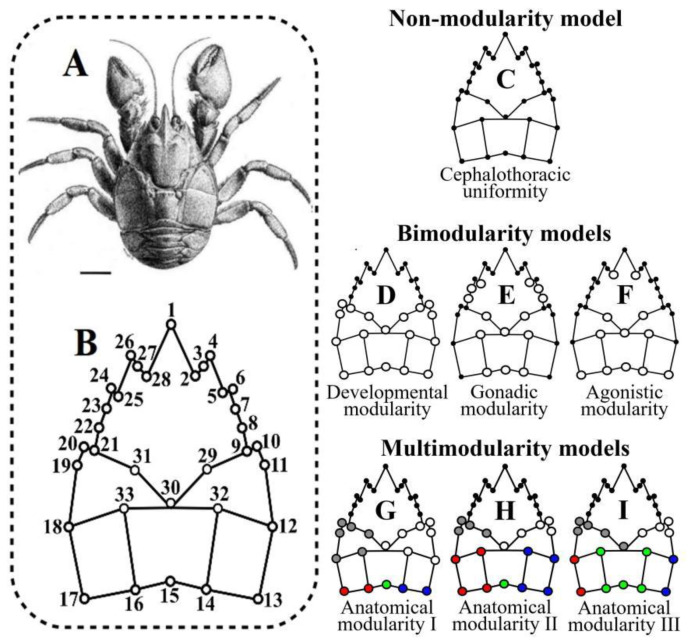
Morphological analysis applied to the dorsal cephalothoracic surface of *Aegla araucaniensis.* (**A**) Paratype of *A. araucaniensis* (bar = 5 mm). (**B**) Configuration of landmarks used in this study (landmark names described in Table 1 of Barría et al [14]. (**C**) Simplest model of isotropic covariation among landmarks. (**D**) Three additional bi-modularity models, including the disposition of the developmental modules. (**E**) Two functional models associated with the gonadic modularity in females. (**F**) The agonistic modularity of males. (**G**–**I**) Anatomical multi-modularity models, including all relations among the anatomical areas of the dorsal surface of the cephalothorax.

**Figure 3 biology-11-00958-f003:**
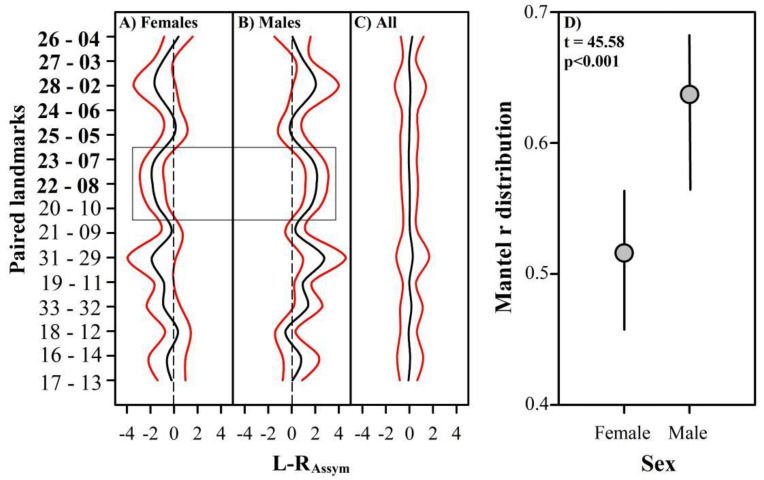
Longitudinal profiles of the directional asymmetry of females (**A**), males (**B**) and the fluctuating asymmetry (**C**) (mean ± 95% CI). Paired landmarks with and without bold represent the cephalic and thoracic modules, respectively. The box includes the paired landmarks that showed significant differences in the asymmetry of males and females. (**D**) Female–male comparison in the distributions of Mantel r obtained by correlating the asymmetric shape components of the developmental modules.

**Figure 4 biology-11-00958-f004:**
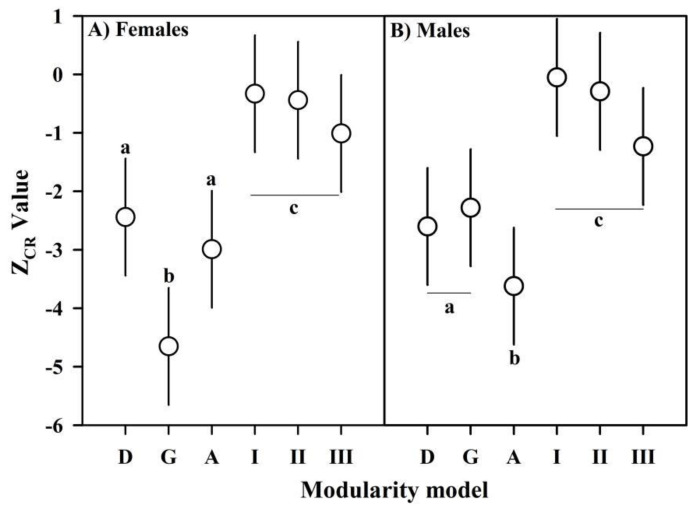
Modular signal based on the CR size effect values (mean Z_CR_ ± 95% CI) for developmental (D), gonadic (G), agonistic (A) and the three anatomical (I–III) modularity models estimated for females (**A**) and males (**B**). Lowercase letters indicate groups of statistically similar models given the relative differences in the modular signal |Ẑ_12_| and overlap of the Z_CR_ distributions (*p*-value).

**Figure 5 biology-11-00958-f005:**
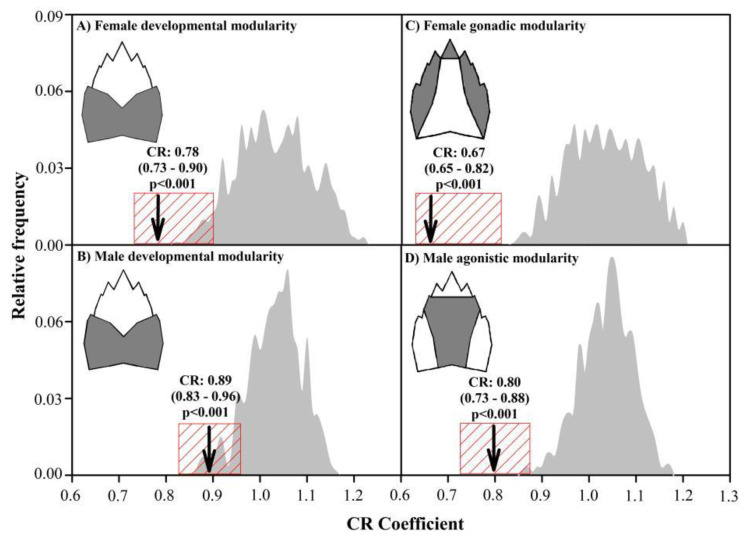
Modularity test showing the variations in the expression of the developmental modularity of females (**A**) and males (**B**) with respect to the best-supported models of gonadic modularity in females (**C**) and agonistic modularity in males (**D**) of *A. araucaniensis*. The mean and distribution of the observed partitions (arrow and red box, respectively) were contrasted with a distribution of arbitrary partitions (grey curves).

**Table 1 biology-11-00958-t001:** Output of two-way Procrustes ANOVA applied to the symmetric and asymmetric shape components. Sex was nested in the variations of both components. df: degrees of freedom, SS: sum of squares, MS: Mean squares, F: F-ratio, *p*-value: probability of significance estimated by Randomized Residuals in a Permutation Procedure (RRPP).

Source of Variation	df	SS	MS	F	*p*-Value (RRPP)
Among individuals (Symmetry component)	1	0.025	0.025	18.632	<0.0001
Within individuals (Directional asymmetry)	1	0.024	0.024	17.681	<0.0001
Among×Within individual interaction (Fluctuating asymmetry)	1	0.001	0.001	1.019	0.408
Error	408	0.53	0.001		

## Data Availability

Not applicable.

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
