# Peer review of "Evolvability in the Cephalothoracic Structural Complexity of *Aegla araucaniensis* (Crustacea: Decapoda) Determined by a Developmental System with Low Covariational Constraint"

_biology, 2022, doi:10.3390/biology11070958_

Round 1

Reviewer 1 Report

The manuscript “Evolvability in the cephalothoracic structural complexity of Aegla araucaniensis (Crustacea: Decapoda) determined by a developmental system with low covariational constraint” presents an innovative approach for Aeglidae, an evolutionarily and ecologically unique crustacean group.  By using geometric morphometrics, the authors investigated the potential evolvability of the adult cephalothorax through sexually dimorphic expressions of functional modularity.

In general, the manuscript is well written and organized. Some comments are below:

1)       Lines 122-125: The role of cheliped size in agonistic confrontations between males and copulatory behavior is well known, but I am not aware of studies showing that the anterior amplitude and prominence size of the latero-frontal spines in males are also related to males’ fitness. Which studies specifically demonstrated these relationships? Please, cite them since this information is crucial for your hypothesis.

2)       Lines 125-127: Would your hypothesis be “We hypothesize that these divergent adaptive causes also influence sexually dimorphic expression of functional modularity”?

3)       Lines 146-147: Are the animals from a single population? Please, include this information. Since interpopulation variation may occur, using a single population is advisable to not bias the results. Figure 2A is not called out in the text; maybe you could cite it in line 147, after “A. araucaniensis”.

4)       Line 176: Try to call out figure 2 in order (first appears Figure 2B, followed by 2D).

5)       Line 322: Did you mean “… the gonadic modularity in females (Figure 5C) and the agonistic modularity…”?

6)       Line 342: “of A. araucaniensis”.

7)       Line 358: “based on”.

8)       Lines 393-396: Maybe this is a pattern for this crustacean group, which would reflect in the group’s taxonomy, based mainly on male specimens.

Please improve the legends of figures and table:

1)       Figure 1: Although some symbols and abbreviations are widely known by carcinologists, it would be good if their definitions were given in the legend (A1, A2…).

2)       Figure 2: Provide a general caption describing the whole figure before “A) Holotype of…”

3)       Table 1: Include the meaning of the abbreviations in the caption, even if they were described in the manuscript text.

Author Response

Response to Reviewer 1 Comments

  •  Lines 122-125: The role of cheliped size in agonistic confrontations between males and copulatory behavior is well known, but I am not aware of studies showing that the anterior amplitude and prominence size of the latero-frontal spines in males are also related to males’ fitness. Which studies specifically demonstrated these relationships? Please, cite them since this information is crucial for your hypothesis.

Response: We committed to the inclusion of three articles to reinforce the idea

Rufino, M.; P. Abelló & A.B. Yule. 2004. Male and female caparace shape differences in Liocarcinus depurator (Decapoda, Brachyura): and application of geometric morphometric analysis to crustacean. Italian Journal of Zoology. 71: 79-83.

Rufino, M.; P. Abelló & A.B. Yule. 2004. Licoarcinus depurator (Brachyura: Portunidae) using geometric morphometrics and the influence of a digitizing method. Journal of Zoology. 269: 458-465.  

Almerão, M.; G. Bond-Buckup & M. de S. Mendoça Jr. 2010. Mating behavior of Aegla platensis (Crustacea, Anomura, Aeglidae) under laboratory conditions.  Journal of Ethology. 28: 87-94.

However, the co-author responsible for editing the bibliography is out of the country and cannot access to endnote to edit the references before the deadline for resend the document. For this reason, we promise include these articles during the editing of proof.

Reviewer:

  • Lines 125-127: Would your hypothesis be “We hypothesize that these divergent adaptive causes also influence sexually dimorphic expression of functional modularity”?

Response: The correction was considered

Reviewer:

3)       Lines 146-147: Are the animals from a single population? Please, include this information. Since interpopulation variation may occur, using a single population is advisable to not bias the results. Figure 2A is not called out in the text; maybe you could cite it in line 147, after “A. araucaniensis”.

Response: Both orretions were considered

Reviewer:

4)       Line 176: Try to call out figure 2 in order (first appears Figure 2B, followed by 2D).

Response: We remove Figure 2B to avoid confusions

Reviewer:

5) Line 322: Did you mean “… the gonadic modularity in females (Figure 5C) and the agonistic modularity…”?

Response:

We made a gramatical correction in the paragraph to reduce the confusions  

Reviewer:

6) Line 342: “of A. araucaniensis”.

Response: The indication was Corrected

Reviewer:

  • Line 358: “based on”.

Response: Indication corrected

Reviewer:

  • Lines 393-396: Maybe this is a pattern for this crustacean group, which would reflect in the group’s taxonomy, based mainly on male specimens.

Response: In fact, the predictions inferred by reviewer are being tested in an additional study which analyzes the evolutionary persistence of functional modularity in Aegla genus    

Reviewer

Please improve the legends of figures and table: 

  • Figure 1: Although some symbols and abbreviations are widely known by carcinologists, it would be good if their definitions were given in the legend (A1, A2…).

Response: The suggestion was considered

2)       Figure 2: Provide a general caption describing the whole figure before “A) Holotype of…”

Response: The requirement was applied

Reviewer

3)       Table 1: Include the meaning of the abbreviations in the caption, even if they were described in the manuscript text.

Response: The requirement was applied

Reviewer 2 Report

This research is very interesting and meaningful for developmental modularity in Aegla araucaniensis. The cephalothoracic geometry of Aeglar aucaniensis shows marked sexual dimorphism; its adaptive causes also promote dimorphic varia-tions in the evolvability of developmental modularity. The authors found areas of instability in the varianceof the asymmetry in both developmental modules. The direct pathway presents intermediate levels of canalization in the covariation of developmental modules, although significantly higher in males. The cephalothoracic palimpsest model of decapods allows studying modularity in an explicit evo-devo context. It has theoretical value and practical significance. However, some revision of the MS is needed. My suggested changes and reviewing comments are shown below on your article.

1. Line 146-147, why are the numbers of females and males analyzed inconsistentWhere were the test samples obtained?

2. Figure 2A is not represented in the text, the authors would add it in the text.

3. Please adjust the format of Table 1 to meet publication requirements.

4. The format of the reference paper needs to be further modified, such as some font bold, italic and so on.

5. Add a conclusion part may make your article more integrity.

Author Response

Response to reviewer 2

Reviewer

  1. Line 146-147, why are the numbers of females and males analyzed inconsistent?Where were the test samples obtained?

Response: We modified the redaction to incorporate this details

Reviewer

  1. Figure 2A is not represented in the text, the authors would add it in the text.

Response: This was corrected in the text

Reviewer

  1. Please adjust the format of Table 1 to meet publication requirements.

Response: This was corrected in the text

  1. The format of the reference paper needs to be further modified, such as some font bold, italic and so on.

Response: All corrections were made

Reviewer

  1. Add a conclusion part may make your article more integrity.

Response: This suggestion was incorporated to the text 

Reviewer 3 Report

I have only minor suggestions/corrections.

Line 39: “an anomuran crab”

Lines 50-51: “for studying the developmental causes of evolution of complex structures.”

Line 57: “However, when combining”

Line 70: “However, the response”

Line 76: “During metamorphosis or late embryogenesis”

Lines 78-79: “The pereonic appendages, or maxillipeds, affected by this direct developmental pathway change”

Figure 1 legend: extra space in Abzhanov & Kaufman

Line 117: “freshwater anomuran crab”

Line 147: “of 55 females and 48 males of”

Line 342: “cephalothorax of A. araucaniensis

Line 344: “constitutes”

Line 357: “of the brachyuran crab”

Line 358: “based on”

Line 381: “Additionally, the”

Line 411: “of anomuran crabs”

The following references are missing italics for taxon names:

10, 14, 15, 17, 18, 57

The following references have most of the title words capitalized (and based on the other references, they probably should not):

6, 11, 20, 23-26

Author Response

Response to reviewer 3

Reviewer

Line 39: “an anomuran crab”

Response: Suggestion corrected

Reviewer

Lines 50-51: “for studying the developmental causes of evolution of complex structures.”

Response: Suggestion corrected

Reviewer

Line 57: “However, when combining”

Response: Suggestion corrected

Reviewer

Line 70: “However, the response”

Response: Suggestion corrected

Reviewer

Line 76: “During metamorphosis or late embryogenesis”

Response: Suggestion corrected

Reviewer

Lines 78-79: “The pereonic appendages, or maxillipeds, affected by this direct developmental pathway change”

Response: Suggestion corrected

Reviewer

Figure 1 legend: extra space in Abzhanov & Kaufman

Response: Suggestion corrected

Reviewer

Line 117: “freshwater anomuran crab”

Response: Suggestion corrected

Reviewer

Line 147: “of 55 females and 48 males of”

Response: Suggestion corrected

Reviewer

Line 342: “cephalothorax of A. araucaniensis

Response: Suggestion corrected

Reviewer

Line 344: “constitutes”

Response: Suggestion corrected

Reviewer

Line 357: “of the brachyuran crab”

Response: Suggestion corrected

Reviewer

Line 358: “based on”

Response: Suggestion corrected

Reviewer

Line 381: “Additionally, the”

Response: Suggestion corrected

Reviewer

Line 411: “of anomuran crabs”

Response: Suggestion corrected

Reviewer

The following references are missing italics for taxon names:

10, 14, 15, 17, 18, 57

The following references have most of the title words capitalized (and based on the other references, they probably should not):

6, 11, 20, 23-26

Response: All corrections were made